# A non-abelian analogue of DBI from $T\overline{T}$

**T. Daniel Brennan, Christian Ferko and Savdeep Sethi**

Enrico Fermi Institute & Kadanoff Center for Theoretical Physics,
University of Chicago, Chicago, IL 60637, USA

## Abstract

The Dirac action describes the physics of the Nambu-Goldstone scalars found on branes. The Born-Infeld action defines a non-linear theory of electrodynamics. The combined Dirac-Born-Infeld (DBI) action describes the leading interactions supported on a single D-brane in string theory. We define a non-abelian analogue of DBI using the $T\overline{T}$ deformation in two dimensions. The resulting quantum theory is compatible with maximal supersymmetry and such theories are quite rare.

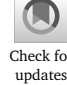

## 1 Introduction

Imagine a $p$-brane embedded in an ambient $(D+1)$-dimensional Minkowski space-time. By definition, any such brane spontaneously breaks the Poincaré symmetry of the ambient space-

time:

$$ISO(D,1) \rightarrow ISO(p,1). \tag{1.1}$$

In particular, the breaking of translational symmetry guarantees the existence of $D-p$ universal scalar fields on the brane world-volume, collectively denoted $\phi$, which are Nambu-Goldstone (NG) bosons for the broken translations. The physics of these modes is governed by the Dirac action,

$$S_{\text{Dirac}} = -T_p \int d^{p+1}\sigma \sqrt{-\det\left(\eta_{\mu\nu} + \partial_\mu\phi\partial_\nu\phi\right)}, \tag{1.2}$$

with brane world-volume coordinates $\sigma$ and a single dimensionful parameter $T_p$. The form of this action is fixed by the broken Lorentz symmetries, which are non-linearly realized. There might also be a function of any additional non-universal scalar fields multiplying this form, which we will not consider here. This action can also equivalently be viewed as the Nambu-Goto action for the brane in static gauge.

The Born-Infeld action, on the other hand, defines a non-linear interacting extension of Maxwell theory with action:

$$S_{\text{BI}} = -T_p \int d^{p+1}\sigma \sqrt{-\det\left(\eta_{\mu\nu} + \alpha F_{\mu\nu}\right)}. \tag{1.3}$$

One of the striking physical differences between Born-Infeld theory and Maxwell theory is the existence of a critical electric field determined by the dimensionful parameter $\alpha$. In string theory, Born-Infeld theory describes the leading interactions for the gauge-field supported on a D-brane [1]. In that context both $T_p$ and $\alpha$ are fixed in terms of the fundamental string tension $\alpha'$ with $\alpha = 2\pi\alpha'$.

The combined Dirac-Born-Infeld (DBI) action is a complete description of the physics of a single D-brane at leading order in string perturbation theory, and under the assumption that acceleration terms like $\partial F$ or $\partial^2 \phi$ are negligible:

$$S_{\text{DBI}} = -T_p \int d^{p+1}\sigma \sqrt{-\det\left(\eta_{\mu\nu} + \partial_\mu\phi\partial_\nu\phi + \alpha F_{\mu\nu}\right)}. \tag{1.4}$$

For multiple coincident branes, the abelian gauge symmetry is replaced by a non-abelian symmetry, and the fields $(\phi, F)$ typically take values in the adjoint representation of the gauge group. We will not assume any particular representation for the scalar fields in this discussion. It is natural to pose the following long considered question: what might replace (1.4) in the non-abelian theory? For the scalar fields appearing in the induced metric of the Dirac action (1.2), one could easily imagine making the replacement

$$\partial_\mu\phi\partial_\nu\phi \rightarrow \text{Tr}\left(D_\mu\phi D_\nu\phi\right), \tag{1.5}$$

where $\phi$ is now matrix-valued and $D$ is an appropriate covariant derivative. For the Born-Infeld action of (1.3), however, an interesting gauge-invariant replacement of this sort is not possible. In (1.5) Tr denotes the trace over gauge indices. When needed, we will use tr to denote the trace over Lorentz indices so that,

$$\text{tr}(F^2) = F_{\mu\nu}F^{\mu\nu}. \tag{1.6}$$

Indeed what one means by the Born-Infeld approximation, namely neglecting acceleration terms like $DF$, is ambiguous. Unlike the abelian case,

$$\left[D_\mu, D_\nu\right] = -iF_{\mu\nu}, \tag{1.7}$$

so there is no clear cut way of truncating the full brane effective action by throwing out acceleration terms.

With considerable hard work there is, however, some data known about brane couplings beyond the two derivative non-abelian kinetic terms $\mathrm{Tr}(F_{\mu\nu}F^{\mu\nu})$ at leading order in string perturbation theory. This information is very nicely summarized in the thesis [2] to which we refer for a more complete discussion. For comparative purposes with our analysis, we note that the known $F^4$ terms are correctly captured by a symmetrized trace prescription [3,4]. Up to overall scaling, they are given by

$$\mathrm{STr}\left(\mathrm{tr}F^4 - \frac{1}{4}(\mathrm{tr}F^2)^2\right), \tag{1.8}$$

with

$$\mathrm{STr}(T_1 T_2 \ldots T_n) = \frac{1}{n!}\sum_{\sigma \in S_n} \mathrm{Tr}(T_{\sigma(1)} T_{\sigma(2)} \ldots T_{\sigma(n)}). \tag{1.9}$$

This prescription is known to fail for higher derivative terms. What is important for us is that (1.8) defines a single trace operator.

To define a non-abelian analogue of the abelian DBI theory (1.4), our approach will be to $T\overline{T}$ deform a non-abelian gauge theory with scalar matter in two dimensions. The $T\overline{T}$ deformation was introduced in [5,6]. In this work, we restrict to bosonic theories for simplicity. A priori this approach has no connection to either brane physics or string theory. We will do this in steps by first recalling known results about deforming free scalars and Maxwell fields [7–9], and then extending to charged matter and non-abelian gauge theories. Other than also involving an infinite collection of irrelevant operators, the reason this approach should be viewed as giving a non-abelian analogue of DBI is that the $T\overline{T}$ deformation of a free scalar with parameter $\lambda$ already gives the Dirac action [7,8]:

$$S_\lambda = \int d^2\sigma \frac{1}{2\lambda}\left(\sqrt{1 + 4\lambda|\partial\phi|^2} - 1\right). \tag{1.10}$$

This direct connection with brane physics is reason enough to suspect that $T\overline{T}$ applied to gauge-theory will give further insight into brane physics.

The $T\overline{T}$ deformation of Maxwell theory, however, is already different from the Born-Infeld theory of (1.3). This is the reason we call the $T\overline{T}$-deformed theory an analogue rather than a generalization of DBI; it does not reduce to DBI even in the case of abelian gauge theory. The couplings are not given by the square-root structure of a relativistic particle but rather by a hypergeometric function [9]. This might not seem very exciting in two dimensions where pure gauge-fields have no propagating degrees of freedom, but that is no longer the case when we add scalar fields, even in the abelian setting. For interesting recent discussions of $T\overline{T}$-deformed gauge theories, see [10,11].

For the non-abelian theory defined using $T\overline{T}$, the $O(F^4)$ terms are already very different from what is known about non-abelian BI theory. Rather than involving a single trace operator like (1.8), they involve double trace operators. The $T\overline{T}$-deformed theory is quite remarkable because it has the following properties:

- The theory is compatible with supersymmetry [12–18]. In fact, if one $T\overline{T}$ deforms a maximally supersymmetric starting theory then this supersymmetry is preserved!

- The theory is believed to exist at the quantum level, unlike DBI which is an effective theory with our present level of understanding.

- The theory has a critical electric field like BI.

This is already quite surprising in the abelian case with uncharged matter. Folklore suggests that some of these properties, like compatibility with maximal supersymmetry, should only have been true for DBI. Indeed there are no obvious reasons that the structures seen here should not emerge from string theory, either in a closed or open string setting. In fact, the $D = 10$ space-time effective action for the type I/heterotic strings does contain a double trace $F^4$ term, which is required for anomaly cancelation in $D = 10$, or more generally required by supersymmetry [19]. In the heterotic string, the term arises at tree-level and takes the schematic form:

$$S_{\text{het}} \sim \int d^{10}x \sqrt{g} e^{-2\phi} \left(\text{Tr}F^2\right)^2 + \dots . \tag{1.11}$$

In the dual type I frame, relevant for a brane picture, the same coupling arises from diagrams with Euler characteristic $-1$ [20, 21]. This leads us to suspect that the $T\overline{T}$ flow equation is connected with corrections to two-dimensional beta functions from higher orders in string perturbation theory.

The paper is organized as follows. Section 2 reviews known results about the $T\overline{T}$ deformation for pure gauge theory; in subsection 2.1 we state the definition of the $T\overline{T}$ flow, in subsection 2.2 we write down the partial differential equation satisfied by the Lagrangian, and in subsections 2.2.1 through 2.2.3 we show three ways of solving this flow equation. In subsection 2.3, we compare the leading irrelevant operators of the $T\overline{T}$ deformed gauge theory to those of Born-Infeld, and compare their corresponding critical electric fields. Section 3 extends these results to the case of gauge theory coupled to scalars: subsections 3.1 through 3.3 use the same three ways of solving the $T\overline{T}$ flow equation to determine the deformed action in this case. The derivation of the flow equations analyzed in these sections is relegated to Appendix A.

## 2 The $T\overline{T}$ Deformation

In this section we will first consider the $T\overline{T}$ deformation of Yang-Mills theory.

### 2.1 The $T\overline{T}$ flow

The $T\overline{T}$-deformation refers to the deformation of a theory by the operator $\det T$. Although this operator is irrelevant, it gives rise to a solvable deformation of the theory which is encapsulated at the classical level in the $T\overline{T}$ flow equation for the deformed Lagrangian $\mathcal{L}_\lambda$:

$$\partial_\lambda \mathcal{L}_\lambda = \det T_{\mu\nu} . \tag{2.1}$$

Because the deformation changes the stress energy tensor itself, this differential equation is self-referential and leads to an infinite series of "corrections" relative to the undeformed theory. Since $2 \times 2$ matrices satisfy the property

$$\det(M) = \frac{1}{2}\left((\text{tr}M)^2 - \text{tr}\left(M^2\right)\right), \tag{2.2}$$

we can write the $T\overline{T}$ flow equation in a manifestly Lorentz-invariant way as

$$\partial_\lambda \mathcal{L}_\lambda = \frac{1}{2}\left(\left(T^\mu_{\ \mu}\right)^2 - T^{\mu\nu}T_{\mu\nu}\right). \tag{2.3}$$

Equation (2.3) will be the starting point for deformations considered here. In this paper we will rely on several techniques to solve for the deformed theory:

- Directly solving the flow equation (2.3) in a series expansion;

- Writing the solution implicitly in terms of a complete integral; and

- Dualizing the field strength $F^2$ to a scalar, deforming, and dualizing back.

## 2.2 Deforming Pure Gauge Theory

Before we go on to solve the $T\overline{T}$ equation to find a non-abelian analogue of the DBI action, we will first illustrate the above techniques by solving for the $T\overline{T}$-deformed Yang-Mills theory. That is, we begin with an undeformed Lagrangian of the form

$$k\mathcal{L}_0 = F^a_{\mu\nu}F^{\mu\nu}_a = \text{Tr}\left(F_{\mu\nu}F^{\mu\nu}\right), \tag{2.4}$$

where we will often suppress the trace for convenience and simply write $F^2$ for $\text{Tr}(F^2)$, so that

$$\mathcal{L}_0 = \frac{1}{k}F^2. \tag{2.5}$$

We retain an overall dimensionless constant $k$ in the Lagrangian; in most of the calculations that follow, which we will suppress factors of $k$ by setting $k = 1$. The value of $k$ does not affect the equations of motion associated with the action (2.5), but the sign of $k$ will be important for determining critical value of the field strength $F^2$. To see the maximum allowed electric field for the deformed theory in Minkowski signature, we will find that we must take $k < 0$ so that the undeformed action is positive (however, all of our other results are valid in either Minkowski or Euclidean signature). We will restore factors of $k$, replacing $F^2 \rightarrow \frac{1}{k}F^2$, when the sign is relevant.

Our goal is to find the deformed Lagrangian $\mathcal{L}(\lambda) = f(\lambda, F^2)$ which solves the flow equation (2.1) with initial condition $\mathcal{L}(0) = \mathcal{L}_0$. Notice that the stress-energy tensor is a single-trace operator in the undeformed theory. In the $\lambda$ expansion, the leading order deformation of the Lagrangian is therefore automatically a double trace operator. Including further corrections in $\lambda$ will only generate higher order multi-trace operators. In particular – as we show in Appendix A – a single-trace deformation like the leading $F^4$ terms of non-abelian DBI in (1.8) will never be generated from a $T\overline{T}$ flow beginning from an undeformed Lagrangian which is only a function of $F^2$.

The details of the calculation of the stress tensor components $T^{(\lambda)}_{\mu\nu}$ for the Lagrangian $\mathcal{L}(\lambda, F^2)$ are presented in Appendix A, where we find the $T\overline{T}$ operator for an arbitrary Lagrangian depending on a field strength $F_{\mu\nu}$ and a complex scalar $\phi$. We can set the scalar $\phi$ to zero in the result of that Appendix to find the flow equation for a pure gauge field Lagrangian. The result, using the shorthand notation $x = F^2$, is

$$\frac{df}{d\lambda} = f(x)^2 - 4f(x)x\frac{\partial f}{\partial x} + 4x^2\left(\frac{\partial f}{\partial x}\right)^2. \tag{2.6}$$

Next we will present several methods for solving (2.6).

### 2.2.1 Series Solution of Flow Equation

The differential equation (2.6) derived in the preceding subsection can be brought into a simpler form by refining our ansatz to

$$f(\lambda, F^2) = F^2 g(\lambda F^2) \tag{2.7}$$

for some new function $g$. For convenience, we define the dimensionless variable $\chi = \lambda F^2 = \lambda x$. Then the function $g$ satisfies the differential equation

$$\frac{\partial g}{\partial \chi} = \left(g(\chi) + 2\chi g'(\chi)\right)^2 . \tag{2.8}$$

One can solve this differential equation by making a series ansatz of the form $g(\chi) = \sum_{n=0}^{\infty} c_n \chi^n$, determining the first several coefficients $c_n$. To order $\chi^6$, the function $g(\chi)$ is given by

$$g(\chi) = 1 + \chi + 3\chi^2 + 13\chi^3 + 68\chi^4 + 399\chi^5 + 2530\chi^6 + \mathcal{O}(\chi^7). \tag{2.9}$$

To determine the generating function, one can refer to an encyclopedia of integer sequences [22] to find that $g$ can be written as a generalized hypergeometric function,

$$g(\chi) = {}_4F_3\left(\frac{1}{2}, \frac{3}{4}, 1, \frac{5}{4}; \frac{4}{3}, \frac{5}{3}, 2; \frac{256}{27} \cdot \chi\right) . \tag{2.10}$$

Thus the full solution for the deformed Lagrangian can be written as

$$\begin{aligned}
\mathcal{L}(\lambda) &= F^2 \cdot {}_4F_3\left(\frac{1}{2}, \frac{3}{4}, 1, \frac{5}{4}; \frac{4}{3}, \frac{5}{3}, 2; \frac{256}{27} \cdot \lambda F^2\right) \\
&= \frac{3}{4\lambda}\left({}_3F_2\left(-\frac{1}{2}, -\frac{1}{4}, \frac{1}{4}; \frac{1}{3}, \frac{2}{3}; \frac{256}{27} \cdot \lambda F^2\right) - 1\right).
\end{aligned} \tag{2.11}$$

The functions on the first and second lines of (2.11) are equivalent because of a hypergeometric functional identity. We will use the expression in the first line, written in terms of ${}_4F_3$ rather than ${}_3F_2$, but we include the second expression to make contact with the work of [9], where this expression was first derived. We also note that the function (2.11) has appeared in an analogue of $T\overline{T}$ defined for $(0+1)$ dimensional theories [23].

### 2.2.2 Implicit Solution

We will also find later that it will be useful to solve the $T\overline{T}$ flow equation using a different method. We begin with the differential equation (2.6), but this time we make the ansatz

$$f(\lambda, F^2) = \frac{1}{\lambda}h(\lambda F^2). \tag{2.12}$$

As before, we define the dimensionless variable $\chi = \lambda F^2$. In terms of $h$, the differential equation becomes

$$4\chi^2\left(h'(\chi)\right)^2 - 4\chi h(\chi)h'(\chi) - \chi h'(\chi) + h(\chi)^2 + h(\chi) = 0. \tag{2.13}$$

Equation (2.13) is quadratic in $h'(\chi)$, so we can solve to find

$$\frac{dh}{d\chi} = \frac{1 + 4h(\chi) - \sqrt{1 - 8h(\chi)}}{8\chi}, \tag{2.14}$$

where we have chosen the root which makes $h'(\chi)$ finite as $\chi \to 0$, assuming $\lim_{\chi \to 0} h(\chi) = 0$.

We may separate variables in (2.14) to write

$$\int \frac{8\,dh}{1 + 4h(\chi) - \sqrt{1 - 8h(\chi)}} = \int \frac{d\chi}{\chi}. \tag{2.15}$$

The integrals can be evaluated in terms of logarithms; exponentiating both sides then yields

$$\chi = C\left(1 - \sqrt{1-8h}\right)\left(3 + \sqrt{1-8h}\right)^3. \tag{2.16}$$

Equation (2.16) implicitly defines the solution $h(\chi)$ to the $T\overline{T}$ flow equation via the roots of an algebraic equation.

   We note that, choosing $C = \frac{1}{256}$, equation (2.16) is consistent with the solution derived in the previous section. Recall that the two ansatzes we made here and in subsection (2.2.1) are related by

$$f(\lambda, F^2) = F^2 f(\chi) = \frac{1}{\lambda} h(\chi), \tag{2.17}$$

so $h(\chi) = \chi f(\chi)$. Indeed, one can check that the function

$$h(\chi) = \chi \cdot {}_4F_3\left(\frac{1}{2}, \frac{3}{4}, 1, \frac{5}{4}; \frac{4}{3}, \frac{5}{3}, 2; \frac{256}{27} \cdot \chi\right), \tag{2.18}$$

satisfies the functional identity

$$\chi = \frac{1}{256}\left(1 - \sqrt{1-8h(\chi)}\right)\left(3 + \sqrt{1-8h(\chi)}\right)^3. \tag{2.19}$$

We therefore see that the hypergeometric (2.11) obtained earlier is, in fact, an algebraic function that can be defined as a root of (2.19).[1]

### 2.2.3   Solution via Dualization

The above result can also be derived in a different way. The details of this procedure do not depend on the sign of our constant $k$ nor the signature, so we will set $k = 1$ and take Minkowski signature for concreteness. The undeformed Lagrangian (2.5) is then

$$\mathcal{L}_0 = \frac{1}{k}F_{\mu\nu}F^{\mu\nu} = -2F_{01}^2, \tag{2.20}$$

which can be equivalently expressed by dualizing the field strength to a scalar, as in

$$\mathcal{L}_0 = \frac{1}{2}\phi^2 + \phi\,\epsilon^{\mu\nu}F_{\mu\nu} = \frac{1}{2}\phi^2 + 2\phi F_{01}. \tag{2.21}$$

The equation of motion for $\phi$ arising from (2.21) is

$$\frac{\delta\mathcal{L}_0}{\delta\phi} = \phi + 2F_{01} = 0, \tag{2.22}$$

so $\phi = -2F_{01}$, and then replacing $\phi$ with its equation of motion yields

$$\mathcal{L}_0 = \frac{1}{2}(-2F_{01})^2 + 2(-2F_{01})F_{01} = -2F_{01}^2, \tag{2.23}$$

which matches (2.20). On the other hand, (2.21) is easy to $T\overline{T}$ deform. After coupling to gravity, one has

$$S_0[g] = \left(\frac{1}{2}\int\sqrt{-g}\,\phi^2\,d^2x\right) + \left(\int\phi\,\epsilon^{\mu\nu}F_{\mu\nu}\,d^2x\right), \tag{2.24}$$

---

[1]Other examples of hypergeometric functions which can be expressed algebraically include those on Schwarz's list [24], which is summarized on Wikipedia.

where the second term is purely topological and thus independent of the metric. The undeformed Lagrangian, then, is a pure potential term $V(\phi) = \phi^2$ for the boson $\phi$. The solution to the $T\overline{T}$ flow equation at finite $\lambda$ for a general potential is well-known [7, 8]; in this case, one finds

$$\mathcal{L}(\lambda) = \frac{\frac{1}{2}\phi^2}{1 - \frac{\lambda}{2}\phi^2} + \phi\,\epsilon^{\mu\nu}F_{\mu\nu} = \frac{\phi^2}{2 - \lambda\phi^2} + 2\phi F_{01}. \tag{2.25}$$

We now integrate out $\phi$. The equation of motion for $\phi$ arising from (2.25) is

$$0 = \frac{\delta\mathcal{L}(\lambda)}{\delta\phi} = 2\phi + F_{01}\left(2 - \lambda\phi^2\right)^2, \tag{2.26}$$

or

$$F_{01} = -\frac{2\phi}{(2 - \lambda\phi^2)^2}. \tag{2.27}$$

To proceed, we series expand (2.27) in $\phi$ to find

$$F_{01} = -\frac{\phi}{2} - \frac{\lambda\phi^3}{2} - \frac{3\lambda^2\phi^5}{8} - \frac{\lambda^3\phi^7}{4} - \frac{5\lambda^4\phi^9}{32} + \mathcal{O}\left(\phi^{11}\right), \tag{2.28}$$

and then apply the Lagrange inversion theorem to find a series expansion for $\phi$ in terms of $F_{01}$, yielding

$$\phi = -2F_{01} + 8\lambda F_{01}^2 - 72\lambda^2 F_{01}^5 + 832\lambda^3 F_{01}^7 - 10880\lambda^4 F_{01}^9 + \mathcal{O}\left(F_{01}^{11}\right). \tag{2.29}$$

Substituting the expansion (2.29) into the action (2.25), and expressing the result in terms of $F^2 = F_{\mu\nu}F^{\mu\nu} = -2F_{01}^2$, gives

$$\mathcal{L}(\lambda) = F^2\left(1 + \lambda F^2 + 3\lambda^2 F^4 + 13\lambda^3 F^6 + 68\lambda^4 F^8 + \cdots\right). \tag{2.30}$$

The Taylor coefficients appearing in (2.30) are precisely those of the hypergeometric (2.10). The procedure of iteratively solving (2.27) for $\phi$ and substituting into (2.25), therefore, reproduces

$$\mathcal{L}(\lambda) = F^2 \cdot {}_4F_3\left(\frac{1}{2}, \frac{3}{4}, 1, \frac{5}{4}; \frac{4}{3}, \frac{5}{3}, 2; \frac{256}{27} \cdot \lambda F^2\right), \tag{2.31}$$

which matches the solution which we derived by different methods above.

This procedure – dualizing the field strength $F^2$ to a scalar $\phi$, $T\overline{T}$ deforming the scalar action, and then dualizing back – is closely related to an observation made in [9]. There the authors noted that, although the deformed Lagrangian (2.11) is quite complicated, the corresponding Hamiltonian satisfies the simple relation

$$\mathcal{H}_\lambda = \frac{\mathcal{H}_0}{1 - \lambda\mathcal{H}_0}, \tag{2.32}$$

where the Hamiltonian is a function of the conjugate momentum

$$\Pi^1 = \frac{\partial\mathcal{L}_\lambda}{\partial\dot{A}_1}. \tag{2.33}$$

The Legendre transform which converts the Lagrangian $\mathcal{L}_\lambda$ to the Hamiltonian $\mathcal{H}_\lambda$ is mathematically equivalent to the process of dualizing the field strength $F_{01}$ to a scalar $\phi$.

### 2.3 Comparison to Born-Infeld

For the moment, we specialize to the abelian case where the Born-Infeld action can be unambiguously defined. The Lagrangian (2.11) differs from the Born-Infeld action in two dimensions. To order $\lambda^4$, our solution has the series expansion

$$\mathcal{L}(\lambda) = F^2 + \lambda F^4 + 3\lambda^2 F^6 + 13\lambda^3 F^8 + 68\lambda^4 F^{10} + \mathcal{O}(\lambda^5). \tag{2.34}$$

On the other hand, the Born-Infeld action (after normalizing the coefficient of $F^2$ to match (2.34) at order $F^2$) has the expansion

$$\frac{1}{2\lambda}\sqrt{1 + 4\lambda F^2} = \frac{1}{2\lambda} + F^2 - \lambda F^4 + 2\lambda^2 F^6 - 5\lambda^3 F^8 + 14\lambda^4 F^{10} + \mathcal{O}(\lambda^5). \tag{2.35}$$

Although the Taylor coefficients for the Born-Infeld action and the "hypergeometric action" differ, both exhibit a critical value for the electric field. In the case of Born-Infeld, this is obvious; replacing $F^2 = -2F_{01}^2$, we see that the action

$$\frac{1}{2\lambda}\sqrt{1 - 8\lambda F_{01}^2} \tag{2.36}$$

is only real for

$$F_{01} < \frac{1}{\sqrt{8\lambda}}. \tag{2.37}$$

To see the critical electric field for the action $\mathcal{L}(\lambda)$ defined in (2.10), it is most convenient to use the implicit form (2.19):

$$\lambda F^2 = \frac{1}{256}\left(1 - \sqrt{1 - 8\lambda\mathcal{L}(\lambda)}\right)\left(3 + \sqrt{1 - 8\lambda\mathcal{L}(\lambda)}\right)^3. \tag{2.38}$$

The right side is maximized when $\mathcal{L}(\lambda) = \frac{1}{8\lambda}$, where it takes the value $\frac{27}{256}$, which means that

$$F^2 < \frac{27}{256\lambda}. \tag{2.39}$$

Recall that our Lagrangian (2.5) contained an overall constant to track signs; to restore factors of $k$, we replace $F^2 \to \frac{1}{k}F^2$. In Minkowski signature, we should take $k < 0$ so that $\mathcal{L}_0 = -\frac{1}{k}F_{\mu\nu}F^{\mu\nu} = -\frac{2}{k}F_{01}^2$ is positive. Letting $k = -1$, we find

$$F_{01} < \sqrt{\frac{27}{512\lambda}}, \tag{2.40}$$

which is a different critical value for the electric field than (2.37).

However, pure Yang-Mills theory in two dimensions has no propagating degrees of freedom, so the difference between the expansions (2.34) and (2.35) does not have much physical effect (at least in infinite volume). To detect the difference between these theories, we should couple the gauge field to matter, as we do in section 3.

## 3 Non-Abelian Analogue of DBI

In this section, we will consider an action for a Yang-Mills gauge field $F_{\mu\nu}^a$ coupled to a scalar $\phi$ in some representation of the gauge group. The undeformed Lagrangian is taken to be

$$\mathcal{L}_0 = F_{\mu\nu}^a F_a^{\mu\nu} + |D\phi|^2 \equiv F^2 + |D\phi|^2, \tag{3.1}$$

where we again use the shorthand $F^2 = \mathrm{Tr}\left(F_{\mu\nu}F^{\mu\nu}\right)$. We have set the overall constant $k$, which appears in (2.5), equal to 1 because we will not analyze critical field strengths in the deformed coupled model. If one were to carry out this analysis, however, one would need an overall minus sign in (3.1) in Minkowski signature.

In what follows, we will also define $x = F^2$ and $y = |D\phi|^2$ for convenience; here

$$
\begin{aligned}
|D\phi|^2 &= \left(D_\mu \phi\right)\left(D^\mu \phi\right)^*, \\
D_\mu &= \partial_\mu - iA_\mu,
\end{aligned}
\tag{3.2}
$$

and gauge group indices will be suppressed.

At finite $\lambda$, we take a general ansatz of the form

$$
\mathcal{L}_\lambda = f(\lambda, x = F^2, y = |D\phi|^2).
\tag{3.3}
$$

The stress tensor components $T_{\mu\nu}^{(\lambda)}$ for the Lagrangian (3.3), and the differential equation arising from (2.3), are worked out in Appendix A. The resulting partial differential equation, equation (A.7), is copied here for convenience:

$$
\frac{df}{d\lambda} = f^2 - 4fx\frac{\partial f}{\partial x} - 2fy\frac{\partial f}{\partial y} + 4x^2\left(\frac{\partial f}{\partial x}\right)^2 + 4xy\frac{\partial f}{\partial x}\frac{\partial f}{\partial y}.
\tag{3.4}
$$

Our goal in the following subsections will be to solve (A.7) by several methods, just as we did in the case of pure gauge theory.

## 3.1 Series Solution of Flow Equation

We know that (A.7) reduces to the Dirac action, (A.9), when the gauge field is set to zero, and that it reduces to the hypergeometric action of Section 2, (A.11), when the scalar is set to zero. Therefore, in the coupled case it is natural to make an ansatz of the form

$$
\begin{aligned}
f(\lambda, x, y) = {}&\frac{1}{2\lambda}\left(\sqrt{1+4\lambda y}-1\right) + {}_3F_4\left(\frac{1}{2},\frac{3}{4},1,\frac{5}{4};\frac{4}{3},\frac{5}{3},2;\frac{256}{27}\cdot\lambda x\right)\cdot x \\
&+ \sum_{n=3}^{\infty}\sum_{k=1}^{n} c_{n,k}\lambda^{n-1}x^k y^{n-k}.
\end{aligned}
\tag{3.5}
$$

The sum on the final line of (3.5) allows for all possible couplings between $F^2$ and $|D\phi|^2$, with the appropriate power of $\lambda$ required by dimensional analysis. One can then determine the coefficients $c_{n,k}$ by plugging the ansatz (3.5) into (A.7) and solving order-by-order in $\lambda$. The result, up to coupled terms of order $\lambda^8$, is

$$
\begin{aligned}
f(\lambda, x, y) = {}&\frac{1}{4\lambda}\left(\sqrt{1+16\lambda y}-1\right) + {}_3F_4\left(\frac{1}{2},\frac{3}{4},1,\frac{5}{4};\frac{4}{3},\frac{5}{3},2;\frac{256}{27}\cdot\lambda x\right)\cdot x \\
&- \lambda^2 x y^2 + \lambda^3\left(4\,xy^3 - 4\,x^2 y^2\right) + \lambda^4\left(18\,x^2 y^3 - 22\,x^3 y^2 - 14\,x y^4\right) \\
&+ \lambda^5\left(-140\,x^4 y^2 + 104\,x^3 y^3 - 65\,x^2 y^4 + 48\,x y^5\right) \\
&+ \lambda^6\left(-165\,x y^6 + 220\,x^2 y^5 - 364\,x^3 y^4 + 680\,x^4 y^3 - 969\,x^5 y^2\right) \\
&+ \lambda^7\left(572\,x y^7 - 726\,x^2 y^6 + 1120\,x^3 y^5 - 2244\,x^4 y^4 + 4788\,x^5 y^3 - 7084\,x^6 y^2\right) \\
&+ \lambda^8\Big(-2002\,x y^8 + 2392\,x^2 y^7 - 3160\,x^3 y^6 + 5814\,x^4 y^5 - 14630\,x^5 y^4 \\
&\qquad\quad + 35420\,x^6 y^3 - 53820\,x^7 y^2\Big).
\end{aligned}
\tag{3.6}
$$

We were unable to find an closed-form expression for the function which generates the couplings (3.6). However, it is interesting to study the corrections in various approximations.

For instance, consider the coupled terms between $F^2$ and $|D\phi|^2$ to leading order in the variable $y = |D\phi|^2$. Retaining only the couplings in (3.6) proportional to $y^2$, one finds

$$f(\lambda, x, y) = \frac{1}{4\lambda}\left(\sqrt{1 + 16\lambda y} - 1\right) + {}_3F_4\left(\frac{1}{2}, \frac{3}{4}, 1, \frac{5}{4}; \frac{4}{3}, \frac{5}{3}, 2; \frac{256}{27} \cdot \lambda x\right) \cdot x - \lambda^2 x y^2 - 4\lambda^3 x^2 y^2$$
$$- 2\lambda^4 x^3 y^2 - 140\lambda^5 x^4 y^2 - 969\lambda^6 x^5 y^2 - 7084\lambda^7 x^6 y^2 - 53820\lambda^8 x^7 y^2$$
$$+ \mathcal{O}\left(\lambda^9, \lambda^3 x y^3\right). \tag{3.7}$$

These series coefficients resum into another hypergeometric function [25],

$$f(\lambda, x, y) = \frac{1}{4\lambda}\left(\sqrt{1 + 16\lambda y} - 1\right) + {}_3F_4\left(\frac{1}{2}, \frac{3}{4}, 1, \frac{5}{4}; \frac{4}{3}, \frac{5}{3}, 2; \frac{256}{27} \cdot \lambda x\right) \cdot x$$
$$- \lambda^2 x y^2 \, {}_3F_2\left(\frac{1}{4}, \frac{1}{2}, \frac{3}{4}; \frac{2}{3}, \frac{4}{3}; \frac{256}{27} \cdot \lambda x\right) + \mathcal{O}\left(\lambda^3 x y^3\right). \tag{3.8}$$

Defining the hypergeometric appearing in the correction term as

$$g(\chi) = {}_3F_2\left(\frac{1}{4}, \frac{1}{2}, \frac{3}{4}; \frac{2}{3}, \frac{4}{3}; \frac{256}{27} \cdot \chi\right), \tag{3.9}$$

one can show that $g$ satisfies the functional relation

$$\chi = \frac{g(\chi) - 1}{g(\chi)^4}. \tag{3.10}$$

The maximum of the function $\frac{g-1}{g^4}$ occurs when $g = \frac{4}{3}$, at which this function takes the maximal value of $\frac{27}{256}$. Therefore, the maximum value of $\chi$ for which the function (3.10) is defined is $\chi = \frac{27}{256}$, giving a critical field strength

$$F^2 < \frac{27}{256\lambda}. \tag{3.11}$$

We note that this is the same value of the critical electric field as that in the uncoupled term involving ${}_3F_4$ in (3.8), which we saw in (2.40) for the case of pure gauge theory. It is reasonable to expect that the value of the critical electric field is modified if one includes corrections to all orders in $|D\phi|^2$, as is the case for the Dirac-Born-Infeld action.

## 3.2 Implicit Solution

We can instead solve (A.7) in terms of a complete integral. First, we refine our ansatz to

$$f(\lambda) = \frac{1}{\lambda} g(\chi, \eta) \quad , \quad \chi = \lambda x \, , \quad \eta = \lambda y \, . \tag{3.12}$$

After doing this, the differential equation becomes

$$0 = 4\chi^2\left(\frac{\partial g}{\partial \chi}\right)^2 + 4\eta\chi \frac{\partial g}{\partial \chi}\frac{\partial g}{\partial \eta} - 4\chi g\frac{\partial g}{\partial \chi} - \chi\frac{\partial g}{\partial \chi} - 2\eta g\frac{\partial g}{\partial \eta} - \eta\frac{\partial g}{\partial \eta} + g^2 + g. \tag{3.13}$$

Making a change of variables to

$$p = \log(\chi) \quad , \quad q = \log(\eta), \tag{3.14}$$

and writing $g(\chi, \eta) = w(p, q)$, the differential equation for $h$ becomes

$$0 = 4\left(\frac{\partial w}{\partial p}\right)^2 + 4\frac{\partial w}{\partial p}\frac{\partial w}{\partial q} - (4w + 1)\frac{\partial w}{\partial p} - (2w + 1)\frac{\partial w}{\partial q} + w^2 + w \, . \tag{3.15}$$

This can be solved by consulting a handbook of partial differential equations (see, for instance, equation 15 in section 2.2.6 of [26]). For any partial differential equation of the form

$$0 = f_1(w)\left(\frac{\partial w}{\partial x}\right)^2 + f_2(w)\frac{\partial w}{\partial x}\frac{\partial w}{\partial y} + f_3(w)\left(\frac{\partial w}{\partial y}\right)^2 + g_1(w)\frac{\partial w}{\partial x} + g_2(w)\frac{\partial w}{\partial y} + h(w), \quad (3.16)$$

the solution $w(x, y)$ is given implicitly by the following complete integral:

$$C_3 = C_1 x + C_2 y + \int \frac{2F(w)\,dw}{G(w) \pm \sqrt{G(w)^2 - 4F(w)h(w)}},$$
$$F(w) = C_1^2 f_1(w) + C_1 C_2 f_2(w) + C_2^2 f_3(w),$$
$$G(w) = C_1 g_1(w) + C_2 g_2(w). \tag{3.17}$$

Our equation (3.15) is precisely of the form (3.16), after identifying the independent variables $p \sim x, q \sim y$, and with the following functions:

$$f_1 = f_2 = 4 \quad, \quad f_3 = 0 \quad, \quad g_1 = -4w - 1,$$
$$g_2 = -2w - 1 \quad, \quad h(w) = w^2 + w. \tag{3.18}$$

Therefore, the functions $F$ and $G$ are

$$F(w) = 4C_1^2 + 4C_1 C_2,$$
$$G(w) = C_1(-4w - 1) + C_2(-2w - 1)$$
$$= (-4C_1 - 2C_2)w - (C_1 + C_2). \tag{3.19}$$

Our solution, then, is

$$C_3 = C_1 p + C_2 q \tag{3.20}$$

$$+ \int \frac{8\left(C_1^2 + C_1 C_2\right) dw}{-(4C_1 + 2C_2)w - (C_1 + C_2) - \sqrt{((4C_1 + 2C_2)w + (C_1 + C_2))^2 - 16\left(C_1^2 + C_1 C_2\right)(w^2 + w)}},$$

where we have taken the negative root in the denominator, appropriate if $C_1 + C_2 < 0$.

Choosing values of the constants $C_1, C_2, C_3$ in (3.21) gives an implicit relation for the function $w(p, q)$ which solves the $T\overline{T}$ flow equation. For instance, if we set $C_2 = 0$ and $C_1 = -1$, equation (3.21) becomes

$$C_3 + \log(\chi) = \int \frac{8\,dw}{4w + 1 - \sqrt{1 - 8w}}, \tag{3.21}$$

which reproduces the result (2.15) which we found in the case of pure gauge theory. In this sense, our implicit solution is a generalization of the technique of section 2.2.2 to the case where $|D\phi|^2 \neq 0$.

The integral appearing in (3.21) can be computed explicitly in terms of logarithms (or, equivalently, inverse hyperbolic tangents). The integral is of the form

$$I(w) = \int \frac{\alpha\,dw}{-\beta w - \gamma - \sqrt{(\beta w + \gamma)^2 - 2\alpha(w^2 + w)}}, \tag{3.22}$$

where

$$\alpha = 8(C_1^2 + C_1 C_2),$$
$$\beta = 4C_1 + 2C_2,$$
$$\gamma = C_1 + C_2. \tag{3.23}$$

The result can be written as

$$
\begin{aligned}
I(w) = \frac{1}{2}\Bigg( & (\gamma - \beta)\tanh^{-1}\left(\frac{\alpha(w+1)+(\beta-\gamma)(\gamma+\beta w)}{(\beta-\gamma)\sqrt{\gamma^2+w^2(2\alpha+\beta^2)+2w(\alpha+\beta\gamma)}}\right) \\
& - \gamma\tanh^{-1}\left(\frac{\gamma^2+w(\alpha+\beta\gamma)}{\gamma\sqrt{\gamma^2+w^2(2\alpha+\beta^2)+2w(\alpha+\beta\gamma)}}\right) \\
& + \sqrt{2\alpha+\beta^2}\tanh^{-1}\left(\frac{\alpha+2\alpha w+\beta(\gamma+\beta w)}{\sqrt{2\alpha+\beta^2}\sqrt{\gamma^2+w^2(2\alpha+\beta^2)+2w(\alpha+\beta\gamma)}}\right) \\
& + (\beta-\gamma)\log(w+1)+\gamma\log(w)\Bigg).
\end{aligned}
$$
(3.24)

Exponentiating both sides then gives

$$
\exp\left(C_3 - C_1 p - C_2 q\right) = \exp\left(I(w)\right).
$$
(3.25)

After simplifying the exponentials of the inverse hyperbolic tangents in (3.25), the right side only involves rational functions and radicals. This relation, therefore, gives an algebraic equation in $w$ whose roots are the solution to the $T\overline{T}$ flow.

By construction, a function $w(p,q)$ which satisfies (3.25) solves the differential equation (3.15). However, this technique is more unwieldy than the direct series solution for generating Taylor coefficients. The main utility of this strategy is the conceptual result that the solution $w(p,q)$ is, in principle, defined by the root of an equation which involves only radicals and quotients, as we saw for the pure gauge theory case in (2.19).

### 3.3  Solution via Dualization

One can also apply the dualization technique of section (2.2) to the coupled action. Begin with the undeformed action

$$
\mathcal{L}_0 = |D\phi|^2 + F^2 \,,
$$
(3.26)

where we put $k = 1$ since the sign will not affect this calculation. Exactly as before, this action is equivalent to

$$
\mathcal{L}_0 = |D\phi|^2 + \frac{1}{2}\chi^2 + \chi\epsilon^{\mu\nu}F_{\mu\nu} \,,
$$
(3.27)

after integrating out the field $\chi$, although this form of the Lagrangian hides some complexity because the covariant derivative $D$ is now non-local in $\chi$. Ignoring this for the moment, we again note that the action coupled to a background metric is of the form

$$
S_0 = \left(\int \sqrt{-g}\,d^2x\,\left(|D\phi|^2+\frac{1}{2}\chi^2\right)\right) + \left(\int d^2x\,\chi\epsilon^{\mu\nu}F_{\mu\nu}\right).
$$
(3.28)

As far as the $T\overline{T}$ deformation is concerned, (3.28) is simply the action of a complex scalar $\phi$ with a constant potential $V = \frac{1}{2}\chi^2$. The solution to the flow equation at finite $\lambda$ is [7,8]

$$
\mathcal{L}(\lambda) = \frac{1}{2\lambda}\sqrt{\frac{(1-\lambda\chi^2)^2}{\left(1-\frac{1}{2}\lambda\chi^2\right)^2}+2\lambda\frac{2|D\phi|^2+\chi^2}{1-\frac{1}{2}\lambda\chi^2}} - \frac{1}{2\lambda}\frac{1-\lambda\chi^2}{1-\frac{1}{2}\lambda\chi^2}+\chi\epsilon^{\mu\nu}F_{\mu\nu}\,.
$$
(3.29)

As in section (2.2), one might hope to iteratively integrate out the auxiliary field $\chi$ in (3.29) in order to express the result in terms of $F^2$. The equation of motion for $\chi$ resulting from (3.29), after solving for $F_{01}$ (and assuming that $\lambda \chi^2 < 2$), is

$$F_{01} = \frac{\chi \left(-|D\phi|^2 \lambda \left(2 - \lambda \chi^2\right) - \sqrt{2|D\phi|^2 \lambda \left(2 - \lambda \chi^2\right) + 1} - 1\right)}{\left(2 - \lambda \chi^2\right)^2 \sqrt{2|D\phi|^2 \lambda \left(2 - \lambda \chi^2\right) + 1}} . \tag{3.30}$$

Solving (3.30) by series inversion to give $\chi$ as a function of $F_{01}$, then substituting back into (3.29), then determines the full $T\overline{T}$ deformed action. The result, up to order $F^8$ and using the shorthand $x = F^2, y = |D\phi|^2$, is

$$\begin{aligned}
\mathcal{L}(\lambda) = {} & \frac{\sqrt{1 + 4\lambda y} - 1}{2\lambda} + \frac{2x \left(\sqrt{1 + 4\lambda y} + 2\lambda y \left(\sqrt{1 + 4\lambda y} + 2\right) + 1\right)}{\left(2\lambda y + \sqrt{1 + 4\lambda y} + 1\right)^2} \\
& + \frac{16\lambda x^2}{\left(2\lambda y + \sqrt{1 + 4\lambda y} + 1\right)^5} \cdot \left[ 2\lambda^3 y^3 \left(3\sqrt{1 + 4\lambda y} + 14\right) + \lambda^2 y^2 \left(17\sqrt{1 + 4\lambda y} + 31\right) \right. \\
& \left. \qquad\qquad + 2\lambda y \left(4\sqrt{1 + 4\lambda y} + 5\right) + \sqrt{1 + 4\lambda y} + 1 \right] \\
& + \frac{128\lambda^2 x^3}{\left(2\lambda y + \sqrt{1 + 4\lambda y} + 1\right)^8} \cdot \left[ 4\lambda^5 y^5 \left(13\sqrt{1 + 4\lambda y} + 96\right) + 2\lambda^4 y^4 \left(183\sqrt{1 + 4\lambda y} + 496\right) \right. \\
& \qquad\qquad + 8\lambda^3 y^3 \left(57\sqrt{1 + 4\lambda y} + 101\right) + 2\lambda^2 y^2 \left(106\sqrt{1 + 4\lambda y} + 145\right) \\
& \left. \qquad\qquad + 6\lambda y \left(7\sqrt{1 + 4\lambda y} + 8\right) + 3\left(\sqrt{1 + 4\lambda y} + 1\right) \right] \\
& + \frac{1024\lambda^3 x^4}{\left(2\lambda y + \sqrt{1 + 4\lambda y} + 1\right)^{11}} \cdot \left[ 2\lambda^7 y^7 \left(323\sqrt{1 + 4\lambda y} + 3266\right) \right. \\
& \qquad\qquad + \lambda^6 y^6 \left(8471\sqrt{1 + 4\lambda y} + 30585\right) + 2\lambda^5 y^5 \left(9879\sqrt{1 + 4\lambda y} + 22795\right) \\
& \qquad\qquad + 4\lambda^4 y^4 \left(4589\sqrt{1 + 4\lambda y} + 8019\right) + 2\lambda^3 y^3 \left(4244\sqrt{1 + 4\lambda y} + 6093\right) \\
& \qquad\qquad + \lambda^2 y^2 \left(2083\sqrt{1 + 4\lambda y} + 2577\right) + 26\lambda y \left(10\sqrt{1 + 4\lambda y} + 11\right) \\
& \left. \qquad\qquad + 13\left(\sqrt{1 + 4\lambda y} + 1\right) \right] . \tag{3.31}
\end{aligned}$$

We have checked by explicit computation that the series expansion (3.31) solves the flow equation (A.7) to order $x^4$.

## Acknowledgements

C. F. and S. S. are supported in part by NSF Grant No. PHY1720480, and C. F. acknowledges support from the divisional MS-PSD program at the University of Chicago. T. D. B. is supported by the Mafalda and Reinhard Oehme Postdoctoral Fellowship in the Enrico Fermi Institute at the University of Chicago. C. F. would like to thank the organizers of the workshop *"New frontiers of integrable deformations"* in Villa Garbald, Castasegna, which provided stimulating discussions related to the subject of this work.

# A Derivation of General $T\overline{T}$ Flow Equation

In this Appendix, we will obtain the flow equation for a sufficiently general Lagrangian for all cases of interest in the main text.

Consider a general $\lambda$-dependent Lagrangian for a complex scalar $\phi$ and field strength $F$:

$$\mathcal{L} = f(\lambda, F^2, |D\phi|^2), \tag{A.1}$$

For convenience, we will also define $x = F^2$ and $y = |D\phi|^2$. As in the main body of the paper, $D$ is the gauge-covariant derivative and the field strength $F$ need not be abelian; we use the shorthand

$$F^2 = F_{\mu\nu}^a F_a^{\mu\nu} = \text{Tr}\left(F^2\right), \tag{A.2}$$

and we will suppress gauge group indices in what follows.

We can now compute the stress-energy tensor by coupling to a background metric and varying with respect to the metric, which gives

$$T_{\mu\nu}^{(\lambda)} = \eta_{\mu\nu}f - 4\frac{\partial f}{\partial x}F_\mu{}^\sigma F_{\sigma\nu} - 2\frac{\partial f}{\partial y}D_\mu\phi D_\nu\overline{\phi} = \eta_{\mu\nu}f - 2\frac{\partial f}{\partial x}\eta_{\mu\nu}F^2 - 2\frac{\partial f}{\partial y}D_\mu\phi D_\nu\overline{\phi}\,, \tag{A.3}$$

where we have used that $F_\mu{}^\sigma F_{\sigma\nu} = \frac{1}{2}\eta_{\mu\nu}\left(F_{\alpha\beta}F^{\alpha\beta}\right)$ in two dimensions.

The determinant of $T$ is then expressed in terms of the combinations

$$T^{\mu\nu}T_{\mu\nu} = \left(\eta^{\mu\nu}f - 2\eta^{\mu\nu}F^2\frac{\partial f}{\partial x} - 2D^\mu\phi D^\nu\overline{\phi}\frac{\partial f}{\partial y}\right)\left(\eta_{\mu\nu}f - 2\eta_{\mu\nu}F^2\frac{\partial f}{\partial x} - 2D_\mu\phi D_\nu\overline{\phi}\frac{\partial f}{\partial y}\right)$$

$$= 2f^2 - 8F^2 f\frac{\partial f}{\partial x} - 4|D\phi|^2 f\frac{\partial f}{\partial y} + 8F^4\left(\frac{\partial f}{\partial x}\right)^2 + 8F^2|D\phi|^2\frac{\partial f}{\partial x}\frac{\partial f}{\partial y} + 4|D\phi|^4\left(\frac{\partial f}{\partial y}\right)^2$$

$$= 2f^2 - 8xf\frac{\partial f}{\partial x} - 4yf\frac{\partial f}{\partial y} + 8x^2\left(\frac{\partial f}{\partial x}\right)^2 + 8xy\frac{\partial f}{\partial x}\frac{\partial f}{\partial y} + 4y^2\left(\frac{\partial f}{\partial y}\right)^2\,, \tag{A.4}$$

and

$$\left(T^\mu{}_\mu\right)^2 = \left(2f - 4F^2\lambda\frac{\partial f}{\partial x} - 2|D\phi|^2\frac{\partial f}{\partial y}\right)^2$$

$$= 4f^2 - 16F^2 f\frac{\partial f}{\partial x} - 8|D\phi|^2 f\frac{\partial f}{\partial y} + 16F^4\left(\frac{\partial f}{\partial x}\right)^2$$

$$+ 4|D\phi|^4\left(\frac{\partial f}{\partial y}\right)^2 + 16F^2\frac{\partial f}{\partial x}\frac{\partial f}{\partial y}|D\phi|^2$$

$$= 4f^2 - 16xf\frac{\partial f}{\partial x} - 8yf\frac{\partial f}{\partial y} + 16x^2\left(\frac{\partial f}{\partial x}\right)^2 + 16xy\frac{\partial f}{\partial x}\frac{\partial f}{\partial y} + 4y^2\left(\frac{\partial f}{\partial y}\right)^2\,. \tag{A.5}$$

Using these, we can write the $T\overline{T}$ operator as

$$\det(T) = \frac{1}{2}\left(\left(T^\mu{}_\mu\right)^2 - T^{\mu\nu}T_{\mu\nu}\right)$$

$$= f^2 - 4fx\frac{\partial f}{\partial x} - 2fy\frac{\partial f}{\partial y} + 4x^2\left(\frac{\partial f}{\partial x}\right)^2 + 4xy\frac{\partial f}{\partial x}\frac{\partial f}{\partial y}\,, \tag{A.6}$$

and hence the $T\overline{T}$-flow equation as

$$\frac{df}{d\lambda} = f^2 - 4fx\frac{\partial f}{\partial x} - 2fy\frac{\partial f}{\partial y} + 4x^2\left(\frac{\partial f}{\partial x}\right)^2 + 4xy\frac{\partial f}{\partial x}\frac{\partial f}{\partial y}\,. \tag{A.7}$$

This is the main differential equation of interest which we will study in the body of this paper. Also we have used $\eta_{\mu\nu}$ for the metric, these results are valid either in Minkowski signature or in Euclidean signature – replacing $\eta_{\mu\nu}$ with $\delta_{\mu\nu}$ in the intermediate steps of these calculations does not affect our final result (A.7).

In the case where we turn off the field strength, setting $x = 0$, this differential equation becomes

$$\frac{df}{d\lambda} = f^2 - 2f\, y \frac{\partial f}{\partial y}\,. \tag{A.8}$$

Imposing the boundary condition that $f(\lambda = 0) = |D\phi|^2$, we find

$$f(\lambda, |D\phi|^2) = \frac{1}{2\lambda}\left(\sqrt{1 + 4\lambda|D\phi|^2} - 1\right)\,. \tag{A.9}$$

On the other hand, in the case where we turn off the scalars (setting $|D\phi|^2 = 0$), the differential equation (A.7) becomes

$$\frac{df}{d\lambda} = f^2 - 4f\, x \frac{\partial f}{\partial x} + 4x^2 \left(\frac{\partial f}{\partial x}\right)^2\,, \tag{A.10}$$

which has the solution

$$f(\lambda, F^2) = F^2 \cdot {}_3F_4\left(\frac{1}{2}, \frac{3}{4}, 1, \frac{5}{4}; \frac{4}{3}, \frac{5}{3}, 2; \frac{256}{27} \cdot \lambda F^2\right)\,. \tag{A.11}$$

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
