# Peer review of "A Non-Abelian Analogue of DBI from $T \overline{T}$"

_SciPost Physics, doi:SciPost Phys. 8, 052 (2020)_

## Round 2 · Referee Report · Anonymous · 2020-3-22

Report
The paper studies the lowest TTbar deformation of YM theory and of scalar QCD in 2D. The results are
the solution of the deformation equations in these cases in closed form. These are then compared with DBI action and the existence of a similar bound for the electric field is underlined. The two actions are of course different.
The computations are clear and well explained, the explicit results are original and are given in terms of generalised hypergeometric functions. Their derivation is fully motivated.
I think the authors should have done a step forward in the analysis by bringing the deformation to the partition function and show how it works at that level.
In particular, YM2 is a topological QFT and there are famous elegant formulas for its partition function. Is the TTbar deformed theory still topological? How is its partition function related to the YM2 one? Is there an explicit form for it on the torus?
I think that answering the above questions would improve the quality of the paper. Therefore
I suggest that the authors could enlarge the focus by discussing the partition function of the deformed theories in an additional section, before the paper gets published.
Author: Christian Ferko on 2020-03-23 [id 776]
(in reply to Report 1 on 2020-03-22)
We thank the referee for his or her comments.
In response to the suggestion that we include a discussion of the $T \overline{T}$ deformed $\text{YM}_2$ partition function, we would like to point out that this has been done in an earlier work, arXiv:1912.04686 (our reference [11]). There the authors derive a flow equation for the $\text{YM}_2$ partition function on an arbitrary background metric and interpret its solution as a deformation of the quadratic Casimir eigenvalues associated with representations of the gauge group.
Although this preceding work nicely describes the partition function for pure Yang-Mills theory, our work focuses on $\text{YM}_2$ coupled to scalars. In this case, the undeformed partition function is not topological and has not been written down in closed form. Nonetheless, the deformed partition function for this case will satisfy the usual flow equation for $T \overline{T}$ deformed partition functions; we chose not to study it here since it is complicated even in the $\lambda = 0$ limit.
Anonymous on 2020-03-24 [id 777]
Thanks for the clarification about reference [11]. I see that your formulas then makes contact with their results via your ref [9]. I guess you could underline the earlier progresses in [11] and address the possible extensions to scalar QCD2. This at least in the large N limit or from the perturbative view point in the gauge coupling.

---

## Editorial Decision

published